# Hydration and Mechanical Properties of Calcium Sulphoaluminate Cement Containing Calcium Carbonate and Gypsum under NaCl Solutions

**DOI:** 10.3390/ma15030816

**Published:** 2022-01-21

**Authors:** Jianzheng Wang, Shilin Song, Yu Zhang, Tao Xing, Ying Ma, Haiyan Qian

**Affiliations:** 1College of Materials Science and Engineering, Nanjing Tech University, Nanjing 211816, China; 201961203182@njtech.edu.cn (J.W.); 202061103080@njtech.edu.cn (S.S.); 201861103080@njtech.edu.cn (T.X.); maying@njtech.edu.cn (Y.M.); 2State Key Laboratory of Materials-Oriented Chemical Engineering, Nanjing Tech University, Nanjing 211816, China; 3Quechen Silicon Chemical Co., Ltd., Wuxi 214196, China

**Keywords:** calcium sulphoaluminate cement, calcium carbonate, gypsum, hydration, compressive strength, NaCl solutions

## Abstract

Hydration characteristics and mechanical properties of calcium sulphoaluminate (CSA) cement with different contents of CaCO_3_ and gypsum under NaCl solutions were studied, using the testing methods of isothermal calorimetry, X-ray diffraction (XRD), mercury intrusion porosimetry (MIP), linear shrinkage, and compressive strength. Results show that CaCO_3_ can promote hydration and reduce the hydration heat of CSA cement. The reaction between gypsum and C_4_A_3_$\overset{\text{-}}{S}$ releases a large quantity of heat in the initial hydration period; however, over 3 days of accumulation, the level of hydration heat is reduced. Under NaCl solutions, the aluminate phase has difficulty reacting with CaCO_3_ to form carbonate phase but combines with chloride ions to form Friedel’s salt. On the contrary, gypsum reduces aluminate phase, and the content of Friedel’s salt is also reduced. Furthermore, CaCO_3_ and gypsum both increase the total porosity of the CSA cement paste under NaCl solutions during the early curing phase, and over the long-term, pore structure is also optimized. CaCO_3_ and gypsum reduce the linear shrinkage of CSA cement paste under NaCl solutions. Overall, the compressive strength of CSA cement is reduced with the addition of CaCO_3_, and the trend will be sharper with the increase in CaCO_3_. However, when it comes to gypsum, the compressive strength is almost the same during early curing, but in the long-term, compressive strength improves. Essentially, the compressive strength of CSA cement mortar with CaCO_3_ and gypsum will improve under NaCl solutions.

## 1. Introduction

With the exploration of marine resources and the development of the marine industry, requirement for the construction of offshore infrastructure is increasing. During the last two centuries, ordinary Portland cement (OPC) was widely used in marine engineering construction. However, the total amount of energy consumption and carbon dioxide emission during the production of OPC was huge. Moreover, the main hydration products of OPC are Ca(OH)_2_ and C-S-H gel, which easily eroded under marine conditions [1,2]. Herein, demand for new low-carbon cementitious materials with high corrosion resistance is increasing.

Calcium sulphoaluminate (CSA) cement is a kind of cementitious material with low carbon emissions and low energy consumption, as well as good performance in resisting seawater erosion. It has been drawing great interest among industry experts and scientists since the 1970s [3,4,5,6,7]. CSA cement was independently researched and invented by China’s Academy of Building Materials Science in the 1970s and has been investigated and used for over 40 years [8,9]. However, CSA cement was mainly used as a special cementitious material in marine engineering for construction crack compensation or frigid construction, owning to its shortcomings of rapid hardening, low long-term strength, and fewer applications in field construction [10,11].

As is well known, concrete will inevitably erode or be destroyed under marine conditions or high chloride content environments as chloride ions permeate, while the erosion level of concrete is related to factors of permeation time, chloride ion concentration, temperature, and humidity, etc. [12,13,14]. Several studies have shown that AFm phase can combine chloride ions through chemical reactions, while the C-S-H gel combines chloride ions through physical bonding. Herein, the presence of C-S-H gel could delay the infiltration process and reduce the chloride concentration in concrete pore solutions, which is contributes greatly to improving the structural strength of concrete [15,16].

The main minerals in CSA cement are ye’elimite (4CaO·3Al_2_O_3_·SO_3,_ C_4_A_3_$\overset{\text{-}}{S}$ (Cement nomenclature will be used, i.e., SO_3_=$\overset{\text{-}}{S}$, SiO_2_=S, CaO=C, Al_2_O_3_=A, CO_2_=$\overset{\text{-}}{C}$, H_2_O=H)), dicalcium silicate (2CaO·SiO_2_, C_2_S), anhydrite (CaSO_4_), and iron phase. The reaction rate of C_4_A_3_$\overset{\text{-}}{S}$ is very fast, providing high early-stage strength for CSA cement. The hydration of C_2_S is very slow, which could improve the later life strength of CSA cement. CaSO_4_ is mainly added to control the reaction rate of CSA cement and to react with C_4_A_3_$\overset{\text{-}}{S}$ to form ettringite. CaCO_3_ is mainly added to cement as a filler to reduce costs [17]. The chemical reactions of the main minerals in CSA cement are shown:(1)$$C_{4}A_{3}\overset{\text{-}}{S}+18H=C_{4}A\overset{\text{-}}{S}H_{12}(\mathrm{AFm})+2{\mathrm{AH}}_{3},$$
(2)$$C_{4}A_{3}\overset{\text{-}}{S}+2C\overset{\text{-}}{S}+38H=C_{6}A{\overset{\text{-}}{S}}_{3}H_{32}(\mathrm{AFt})+2{\mathrm{AH}}_{3},$$
2C_2_S + 4H = C-S-H + CH,(3)


(Notation with: C = CaO, A = Al_2_O_3_, $\overset{\text{-}}{S}$ = SO_3_, S = SiO_2_, H = H_2_O).

In recent years, several kinds of CaCO_3_ were investigated as possible cement fillers in various studies, such as limestone powder [17,18,19], dolomite powder [20], and coral sand powder [21,22]. Results show that CaCO_3_ can promote the early hydration of cement and can react with C_4_A_3_$\overset{\text{-}}{S}$ to form hemicarbonate (4CaO·Al_2_O_3_·0.5CO_2_·12H_2_O, Hc) and monocarbonate (4CaO·Al_2_O_3_·CO_2_·11H_2_O, Mc), which can stabilize ettringite and refine the pore structure of cement [23,24]. Gypsum can adjust the setting time of cement, control the hydration rate, affect the composition of hydration products, and adjust the strength and expansion capacity of cement [25]. Different types of gypsum play different roles [26,27].

It is hard to ignore the fact that the freshwater and other raw materials required for offshore construction are difficult to attain, because the transportation of freshwater and raw materials is expensive. It follows, that increasing the use of seawater and local materials will shorten construction time and reduce the transportation cost of materials [28,29,30]. Once concrete has been exposed to a chloride ion rich environment over the long term, it will inevitably be destroyed by the infiltrating chloride ion [31,32]. NaCl can accelerate cement hydration in the early stages of hydration. In NaCl solutions, NaCl can accelerate the dissolution of gypsum to form ettringite and increase the heat of hydration [33,34,35]. Meanwhile, NaCl can significantly increase the compressive strength of cement [36]. In this work, the hydration process, compressive strength development and linear expansion rate of CSA cement will be detailed, by illustrating the testing methods of isothermal calorimetry, X-ray diffraction, and pore structure development in the presence of different contents of CaCO_3_ and gypsum under fresh water and NaCl solutions.

## 2. Materials and Methods

### 2.1. Raw Materials

The CSA cement used in this study was a variety of low-alkali CSA cement, which was produced by the Tangshan Liujiu Cement Co., Ltd., Tangshan, China. The CaCO_3_, CaSO_4_·2H_2_O, and NaCl used were all analytical grade (AR) reagents (99.5% content). The chemical composition of CSA cement is shown in Table 1. Figure 1 presents the XRD pattern of CSA cement and the XRD pattern of CaCO_3_ and gypsum are shown in Figure 2. As tested, the main mineral phases of CSA cement are ye’elimite (C_4_A_3_$\overset{\text{-}}{S}$), dicalcium silicate (C_2_S), anhydrite (CaSO_4_), mayenite (12CaO·7Al_2_O_3_, C_12_A_7_), and calcite (CaCO_3_). The mineral phase quantitative analysis of CSA cement is shown in Table 2 using the software X’Pert HighScore Plus version 3.0e.

### 2.2. Sample Preparation

Powder compositions of samples were accurately weighed and well-mixed separately for over 10 h in mixers before they were hydrated with water/solutions. Detailed proportions of the samples are shown in Table 3. Figure 3 shows raw materials being mixed in the mixing machine and the mixtures being stirred with water or solutions in blenders, respectively.

Afterwards, cement pastes of the mixtures were prepared by an overhead stirrer separately with a water-to-cement ratio (w/c) of 0.5 under 600 revolutions per minute. The pastes were stored in sealed vessels at 20 °C. At each curing time, the hydrated pastes were cut into small pieces and soaked in ethanol for 24 h, then vacuum oven dried at 40 °C. Small pieces of the dried samples were used for the test of Mercury intrusion porosimetry (MIP), while the ground fine samples were prepared for the characterization of X-ray diffractometry (XRD).

The paste samples for shrinkage tests were cast into molds with a size of 20 mm × 20 mm × 80 mm at a water/solution to binder ratio (w/b) of 0.35. Mortars of the samples were cast using standard mortars with a proportion of cement:sand:water (solution) = 1:3:0.5 and a prismatic size of 40 mm × 40 mm × 160 mm. Specimens were cured in molds at 20 ± 1 °C and 95% relative humidity (RH) for 24 h. Then, they were demolded and cured in water or a solutions bath until the characterization was performed.

### 2.3. Testing

#### 2.3.1. Isothermal Calorimetry

An 8-channel isothermal calorimeter (TAM Air; Thermometric AB, Stockholms lan, Sweden) was used to record the hydration (exothermic performance) of each sample. A total of 4 g of binder and 2 g of deionized water were mixed in standard plastic bottles. Next, bottles were immediately put in the isothermal calorimeter and the heat flow curve of the samples were recorded at a constant temperature of 20 °C for 72 h.

#### 2.3.2. X-ray Diffraction

Rigaku’s MiniFlex 600 X-ray powder diffractometer (Rigaku Corporation, Tokyo, Japan) with Cu Kα radiation was used to collect the XRD data of raw materials and hydrated dry samples. Meanwhile, software HighScore Plus was used to quantitate the mineral phase compositions of the samples. The scanning was performed over an accelerating voltage of 40 V and a current of 15 mA, with a 2θ range of 5–65° and a 2θ increment of 0.02°/step.

#### 2.3.3. Pore Structure Test

MIP was used to evaluate the pore size distribution of the paste samples. The testing pore size range was 0.007–100 μm with the testing pressure ranging from 1.5–350 kPa to 140–420 kPa. The pore diameters and cumulative pore volumes could be obtained by the MIP curves.

#### 2.3.4. Linear Shrinkage Test

According to JC/T 603-2004, linear shrinkage was measured by testing the length change of prismatic specimens. The specimens were demolded to place in water or a solution bath curing at 20 ± 1 °C during the full testing period. Six specimens were prepared for each sample.

#### 2.3.5. Compressive Strength Test

Compressive strength of the mortar sample was measured with a universal mechanical testing machine when there were 6 specimens to be tested for each mix and the mean value was reported according to EN-196-1-2005.

## 3. Results

### 3.1. Heat Evolution

The heat flow and total cumulated heat curves of CSA cement and the mixed samples with CaCO_3_ or gypsum at a w/c of 0.5 over 72 h are shown in Figure 4 and Figure 5. It is clear that the hydration of CSA cement mixed with CaCO_3_ or gypsum can be divided into five stages [37]: (1) pre-induction period; (2) induction period; (3) acceleration period; (4) retarding period; and (5) stable period.

As shown in Figure 4, there are three exothermic peaks on the hydration heat release curve of CSA cement. The first exothermic peak is related to the dissolution heat release of CSA cement particles [38]. The second exothermic peak refers to the first main exothermic peak and is attributed to the hydration of C_4_A_3_$\overset{\text{-}}{S}$ and the formation of AFt and aluminate hydroxide (AH_3_). The third exothermic peak at around 20 h attributes the hydration of the remaining C_4_A_3_$\overset{\text{-}}{S}$, leading to the formation of monosulphate when all the anhydrite is depleted [39].

CaCO_3_ enhances the third exothermic peak, which means the presence of CaCO_3_ can change the hydration process and effect the hydration product composition of CSA. It is possible that CaCO_3_ provides nucleation sites for hydration, which can promote the further formation of AFt [40]. It is worth noting that Hc or Mc phase can be formed in the presence of CaCO_3_, which brings the possibility of forming a sharper peak for CSA cement with CaCO_3_. Meanwhile, a small amount of CaCO_3_ can increase the total accumulated heat of CSA cement, while a large quantity of CaCO_3_ will show the opposite trend.

In order to compare the hydration procedure of CSA with and without gypsum, CSA with 5 wt% and 10 wt% gypsum was measured (Figure 5). The hydration of CSA was significantly advanced by the presence of gypsum, and the trend is more obvious with the increase in gypsum. The acceleration of the early hydration of CSA corresponded to copious amounts of SO_4_^2−^ promoting the rapid formation of AFt in the very early hydration period [25]. It is assumed that the dissolution exotherm is increased and the reaction of C_4_A_3_$\overset{\text{-}}{S}$ is accelerated in CSA cement containing gypsum. Meanwhile, the third exothermic peak of CSA cement with 10 wt% gypsum disappeared, which means that AFm phase is hard to form in the early hydration period when SO_4_^2-^ is sufficient. As a result, the heat release of CSA cement with extra gypsum is higher than that of CSA in 24 h, while the total accumulated hydration heat of CSA cement is reduced in the presence of gypsum within 72 h.

### 3.2. X-ray Diffraction Analyses and Phasequantitative Analyze by XRD-Rietveld Method

Powder samples of CSA cement were investigated with XRD to identify crystalline phases at different curing stages both in water and NaCl solutions (Figure 6). The diffraction peaks of ettringite, C_4_A_3_$\overset{\text{-}}{S}$, anhydrite and CaCO_3_ were observed at different stages. Ettringite and aluminum hydroxide (Al_2_O_3_·3H_2_O, AH_3_) are the main hydration products of CSA cement, while AH_3_ is mainly gelatinous, and its diffraction peak is not obvious on the XRD pattern. Calcium hydroxide (Ca(OH)_2_, CH) does not appear due to its reaction with AH_3_ and anhydrite to form ettringite so it is consumed in the early curing phase [41], as shown in Equation (4). The diffraction peak of strätlingite is obvious in sample C after curing for 90 days. It should be noted that a small diffraction peak of Friedel’s salt appears around 11.2° within 3 days in C-2, which indicates that chloride ions had been mixed into the paste and chemically combined with AFm phase [42].

The XRD patterns of CSA cement mixed with CaCO_3,_ or gypsum are shown in Figure 7. It can be seen that the Hc is observed and Mc is not shown when CSA is mixed with 5% CaCaO_3_ (Equation (7)) [43]. Meanwhile, Hc phase is still not observed in C. Thus, the diffraction peak intensity of Friedel’s salt in CC5-2 and CS5-2 is lower than that of pure CSA cement (sample C-2). This is owing to the fact that CaCO_3_ and gypsum contribute to the formation of ettringite when AFm phase is reduced. Herein, the formation of Friedel’s salt is inhibited:(4)$$3\mathrm{CH}+3C\overset{\text{-}}{S}+{\mathrm{AH}}_{3}+26H=C_{6}A{\overset{\text{-}}{S}}_{3}H_{32}(\mathrm{AFt}),$$
(5)$$C_{4}A\overset{\text{-}}{S}H_{12}+2{\mathrm{Cl}}^{-}=C_{4}{\mathrm{ACl}}_{2}H_{11}(\mathrm{Friedel}’s\;\mathrm{salt})+{\mathrm{SO}}_{4}2^{-}+2H,$$
2C_2_S + 4AH_3_ + 5H = C_2_ASH_8_, (6)
(7)$$6C_{4}A_{3}\overset{\text{-}}{S}+2C\overset{\text{-}}{C}+135H=2C_{4}A{\overset{\text{-}}{C}}_{0.5}H_{11}(\mathrm{Hc})+2\mathrm{AFt}+14{\mathrm{AH}}_{3}+5\mathrm{CH}$$

(Notation with: C = CaO, A = Al_2_O_3_, $\overset{\text{-}}{S}$ = SO_3_, $\overset{\text{-}}{C}$ = CO_2_, S = SiO_2_, H = H_2_O).

Quantitative calculation of the degree of hydration and content of Friedel’s salt under NaCl solutions is shown in Figure 8. The cement paste combines with chloride ions to generate Friedel’s salt due to the presence of Afm phase in the cement at the initial stage of erosion [44]. With the consumption of Afm phase, the content of Friedel’s salt reaches a stable level gradually. In comparison, the content of Friedel’s salt in C-2 is significantly higher than that of CC5-2 and CC15-2, indicating that the addition of CaCO_3_ reduces the formation of Friedel’s salt. The reason is that CaCO_3_ will reduce the formation of Afm phases. Therein, the contents of Afm phases are reduced and the formation of Friedel’s salt deteriorates. The presence of Gypsum promotes the formation of ettringite and reduces the formation of Afm, which greatly limits the formation of Friedel’s salt. CS15-2 makes this particularly obvious:

### 3.3. Pore Size Distribution

Figure 9 exhibits the pore size distribution and accumulated porosity of pure CSA cement paste and CSA cement pastes with CaCO_3_ and gypsum under NaCl solutions over 180 days. When hydrated for 3 days, the presence of CaCO_3_ and gypsum will promote the formation of larger width pores [45]. As discussed above, CaCO_3_ and gypsum promote the formation of ettringite, then larger sized ettringite crystals will easily form in the early curing phase, which will lead the formation of larger pore spaces. The role of gypsum is particularly obvious [46]. This is the main reason why the porosity of the CSA paste mixed with gypsum is lower than that of CSA samples mixed with CaCO_3_. The presence of CaCO_3_ and gypsum optimizes the pore structure over longer-term curing times. Appropriate amounts of CaCO_3_ and gypsum have a positive effect on the densification of cement pore structure.

Figure 10 shows the total porosity of CSA pastes with CaCO_3_ and gypsum within 180 days of curing in an NaCl solution. Clearly, the total porosity of hydrated paste decreases as curing time increases. The total porosity of CSA samples with CaCO_3_ and gypsum were higher than that of pure CSA cement:

### 3.4. Linear Shrinkage

Linear shrinkage rate curves of the cement pastes within 180 days are shown in Figure 11. Under standard curing conditions, the expansion of CSA cement is owed to the formation of ettringite during the early hydration and hardening process, while the shrinkage is caused by the free water consumption and the reactions of cement in later phases [47]. It was clear that the shrinkage rate of all samples increased and gradually stabilized as curing times extended.

The shrinkage rate of the CSA cement pastes containing CaCO_3_ decreased compared to that of pure CSA cement. The shrinkage rate of CC15 is minimal, which was caused by the CaCO_3_ acting as a filler. CaCO_3_ can also stabilize ettringite and dilute cement [48]. On the contrary, samples containing gypsum had a higher shrinkage rates. This may be related to the later transformation of ettringite [49].

Under NaCl solutions, the drying shrinkage of all samples obviously decreased. The chloride ions, who were invaded in the cement pastes, were chemically combined with AFm phase, which helped the formation and stabilization of ettringite. Accordingly, the shrinkage was reduced. In the early stage of hydration, the shrinkage rate changes greatly due to larger pore sizes. As pore sizes tend to become finer as curing time increases, the shrinkage rate tends to stabilize.

### 3.5. Compressive Strength of Mortar Samples

#### 3.5.1. Compressive Strength of Fresh Water Mixing

Compressive strengths of CSA cement mortars containing CaCO_3_ or gypsum are shown in Figure 12. The compressive strength of all blended mortars increased as the curing time extended. CSA mortars blending with CaCO_3_ had lower compressive strength than that of pure CSA mortars, and the trend seemed to become more obvious by increasing the content of CaCO_3_. CaCO_3_ is mainly used as a filler, it basically does not participate in chemical reaction [40,50]. For CC5, the loss of strength was minimal when CaCO_3_ is used. However, CS5 and CS10 showed higher compressive strength than plain CSA cement mortar. After 3 days, CS10 produced a higher compressive strength than CS5, and a similar strength was reached after 90 days. Gypsum could react with C_4_A_3_$\overset{\text{-}}{S}$ to form ettringite and improve early compressive strength [51].

#### 3.5.2. Compressive Strength of NaCl Solutions Curing

Under NaCl solutions, the compressive strength of all samples increased steadily. CaCO_3_ and gypsum played a similar role in the development of compressive strength compared with freshwater curing. In the presence of chloride ions, the compressive strength increased due to the formation of Friedel’s salt. CS5-2 and CS10-2 reached similar compressive strengths after 90 days. It should be noted that the large amount of ettringite generated inside the mortar sample containing gypsum improved the compressive strength on the one hand, but made the cement mortar more brittle on the other. Although the compressive strength of CC5-2 is lower than that of C-2, the 28-day compressive strength can fully meet the mechanical requirements of 42.5Mpa while decreasing the shrinkage rate.

## 4. Conclusions

This study investigated the influence of CaCO_3_ and gypsum on the hydration and compressive strength of CSA cement under fresh water and NaCl solutions at curing times of up to 180 days. The hydration heat evolution, hydration products and compressive strength were investigated and discussed. Results obtained are as follows:(1)CaCO_3_ and gypsum both accelerate the early hydration of CSA cement and reduce the 3-day accumulate hydration heat.(2)Under NaCl solutions, CaCO_3_ can react with C_4_A_3_$\overset{\text{-}}{S}$ to form Hc, which can combine with Cl^−^ to form Friedel’s salt, while gypsum readily reacts with C_4_A_3_$\overset{\text{-}}{S}$ to form ettringite. Both CaCO_3_ and gypsum can reduce the content of bound chloride ions in CSA cement paste.(3)CaCO_3_ and gypsum will obviously increase the pore size of the cement paste in the early curing stages and the pore size will become finer as the curing time extends. The total porosity of CSA with CaCO_3_ and gypsum is higher than that of CSA cement up to 180 days.(4)The diluting effect of CaCO_3_ reduces the shrinkage rate of cement paste, while the shrinkage rate found by adding gypsum is higher due to the transformation of ettringite. Under NaCl solutions, the shrinkage rates of CSA obviously decreases.(5)The compressive strength of CSA mortars containing CaCO_3_ decreases, while gypsum can increase the compressive strength of mortar samples. Under NaCl solutions, the compressive strength develops better.

## Figures and Tables

**Figure 1 materials-15-00816-f001:**
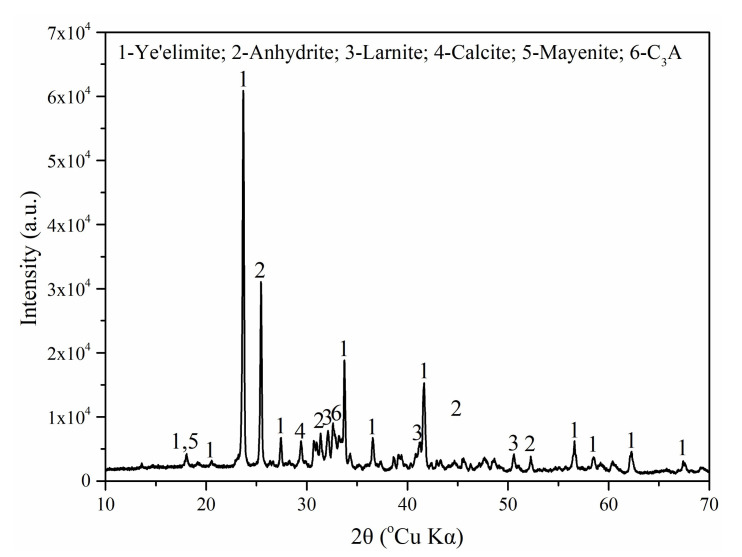
XRD pattern of CSA cement.

**Figure 2 materials-15-00816-f002:**
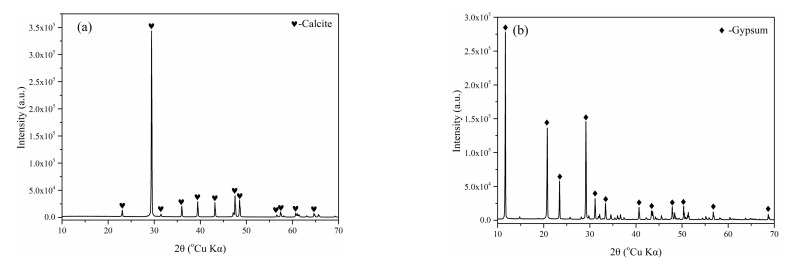
XRD pattern of CaCO_3_ (**a**) and gypsum (**b**).

**Figure 3 materials-15-00816-f003:**
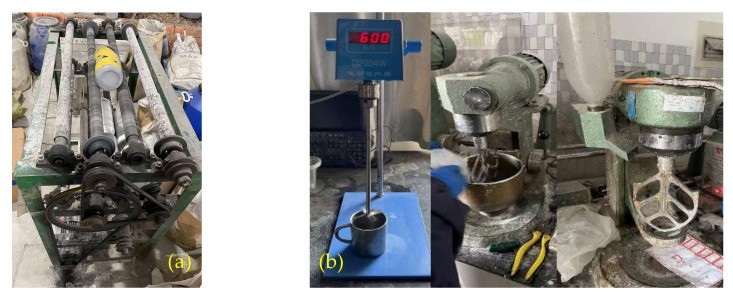
Mixing machine (**a**) and blenders (**b**) making the mixtures in the laboratory.

**Figure 4 materials-15-00816-f004:**
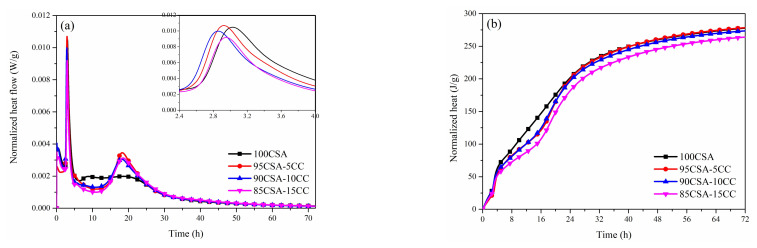
Heat flow curves (**a**) and cumulative heat curves (**b**) of CSA cement mixed with CaCO_3_.

**Figure 5 materials-15-00816-f005:**
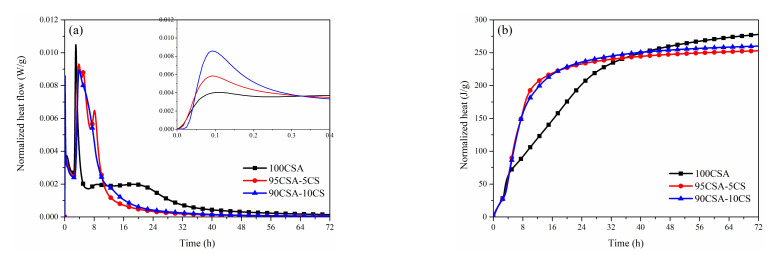
Heat flow curves (**a**) and cumulative heat curves (**b**) of CSA cement mixed with gypsum.

**Figure 6 materials-15-00816-f006:**
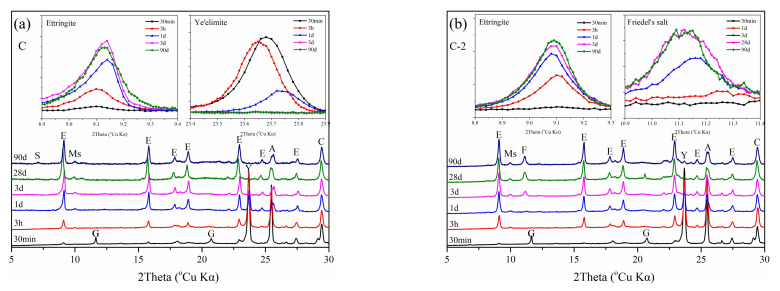
XRD patterns of the CSA cement paste of sample C (**a**) and sample C-2 (**b**) at different curing ages (E—Ettringite, G—Gypsum, Ms—Monosulphoaluminate, Y—Ye’elimite, F—Friedel’s, C—Calcite, A—Anhydrite, S—Strätlingite).

**Figure 7 materials-15-00816-f007:**
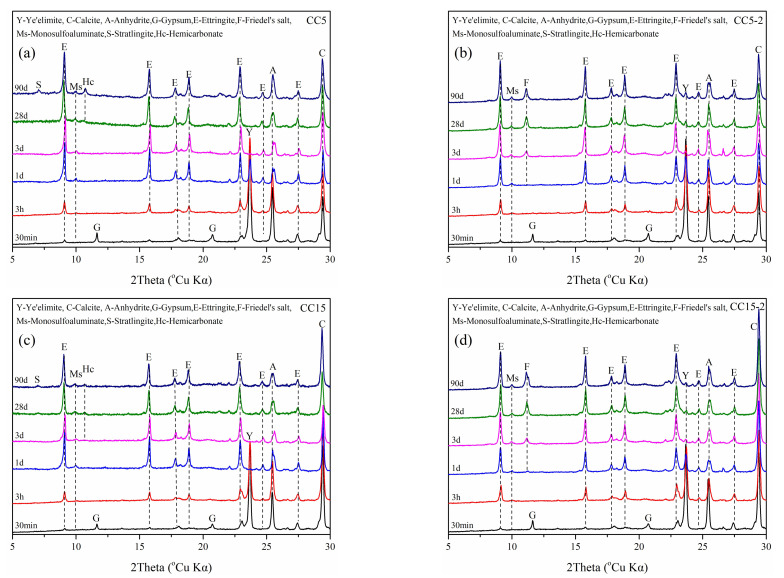
XRD patterns of paste samples of CC5 (**a**), CC5-2 (**b**), CC15 (**c**), CC15-2 (**d**), CS5 (**e**), CS5-2 (**f**), CS15 (**g**) and CS15-2 (**h**).

**Figure 8 materials-15-00816-f008:**
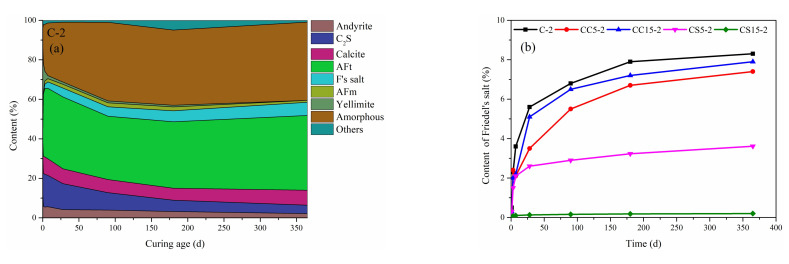
Quantitative calculation of hydration degree (**a**) and Friedel’s salt content (**b**) under NaCl solutions.

**Figure 9 materials-15-00816-f009:**
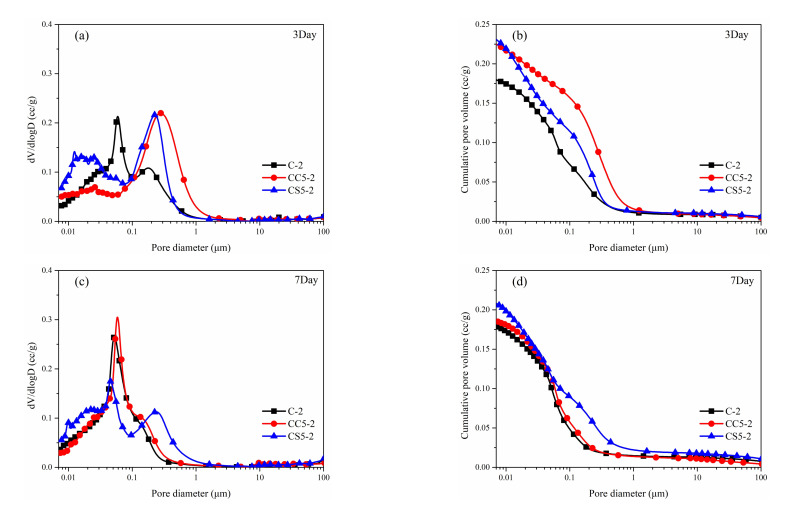
Distribution curve (**a**,**c**,**e**,**g**) and cumulative intrusion curve (**b**,**d**,**f**,**h**) of the hydrated samples at different ages under 3.5% NaCl solutions.

**Figure 10 materials-15-00816-f010:**
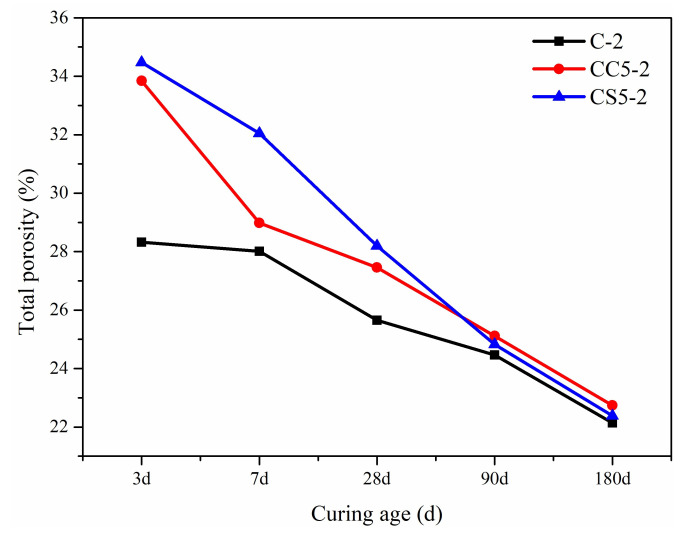
Total porosity development of the hydrated sample up to 180 days.

**Figure 11 materials-15-00816-f011:**
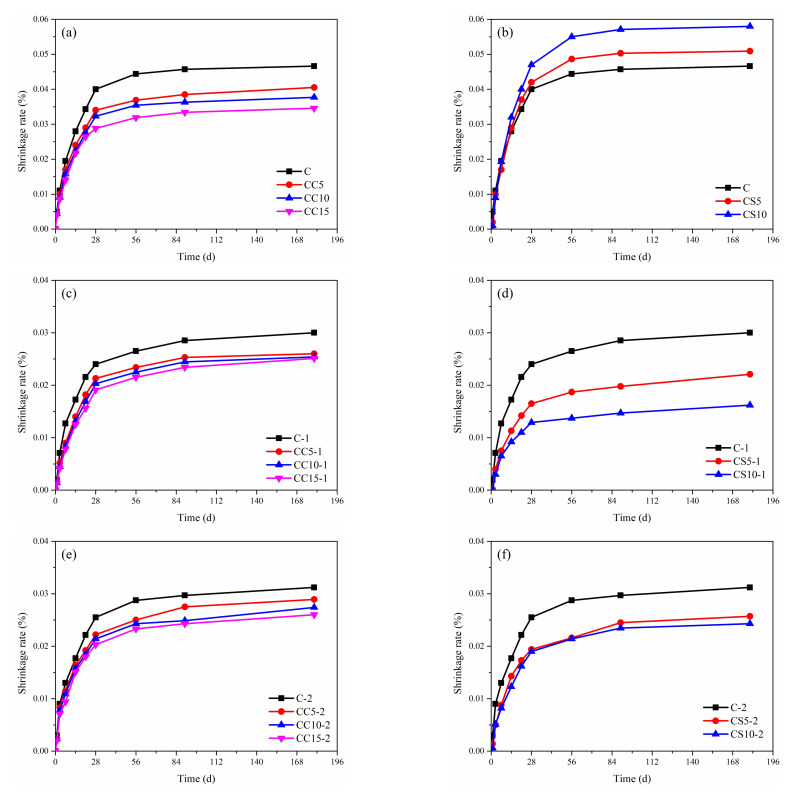
Linear shrinkage rate of paste specimens mixed CaCO_3_ (**a**,**c**,**e**) or gypsum (**b**,**d**,**f**) at different ages.

**Figure 12 materials-15-00816-f012:**
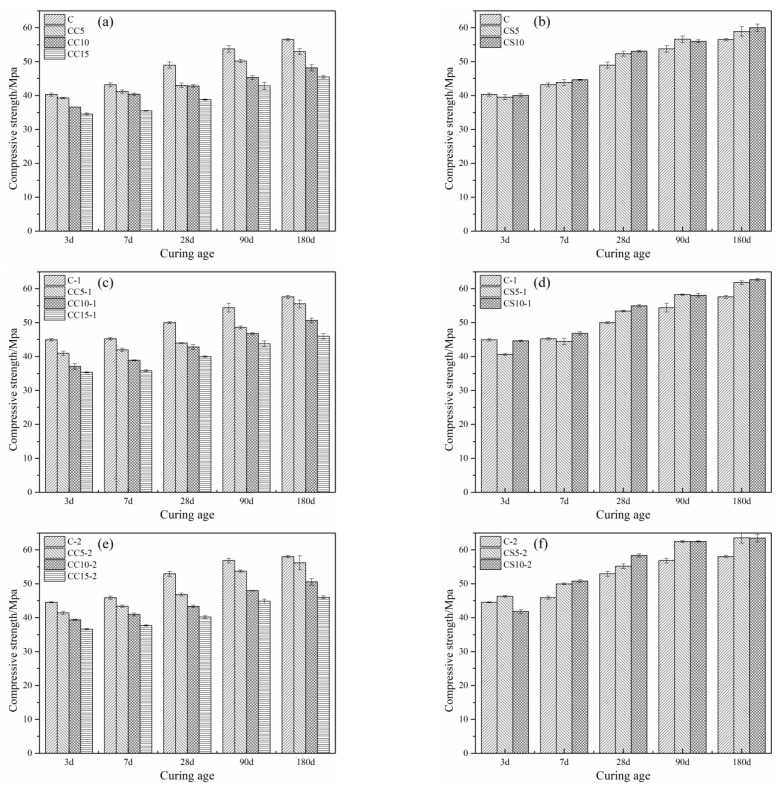
Compressive strength of CSA cement mixed CaCO_3_ (**a**,**c**,**e**) or gypsum (**b**,**d**,**f**) at different ages.

**Table 1 materials-15-00816-t001:** Chemical composition of CSA cement.

Oxide	CSA Cement
Calcium oxide, CaO	42.97
Silicon dioxide, SiO_2_	13.85
Aluminum oxide, Al_2_O_3_	22.14
Sulfur trioxide, SO_3_	11.86
Ferric oxide, Fe_2_O_3_	1.33
Magnesium oxide, MgO	2.86
Potassium oxide, K_2_O	0.31
Sodium oxide, Na_2_O	0.09
Titanium oxide, TiO_2_	0.97
Manganese Oxide, MnO	0.034
Phosphorus Pentoxide, P_2_O_5_	0.16
Loss on ignition, LOI	2.76

**Table 2 materials-15-00816-t002:** Mineral composition of CSA cement.

Mineral	Content/%
$C_{4}A_{3}\overset{\text{-}}{S}$	51.53
C_2_S	21.25
C_3_A	1.71
CaSO_4_	11.52
CaCO_3_	9.81
C1_2_A_7_	1.28

**Table 3 materials-15-00816-t003:** Mixture proportion of all samples (wt%).

Samples	w/c	Cement	CaCO_3_	Gypsum	Water for Mixing	Water for Curing
C	0.5	100	0	0	Deionized water	Deionized water
CC5	0.5	95	5	0	Deionized water	Deionized water
CC10	0.5	90	10	0	Deionized water	Deionized water
CC15	0.5	85	15	0	Deionized water	Deionized water
CS5	0.5	95	0	5	Deionized water	Deionized water
CS10	0.5	90	0	10	Deionized water	Deionized water
C-1	0.5	100	0	0	Deionized water	3.5% NaCl solution
CC5-1	0.5	95	5	0	Deionized water	3.5% NaCl solution
CC10-1	0.5	90	10	0	Deionized water	3.5% NaCl solution
CC15-1	0.5	85	15	0	Deionized water	3.5% NaCl solution
CS5-1	0.5	95	0	5	Deionized water	3.5% NaCl solution
CS10-1	0.5	90	0	10	Deionized water	3.5% NaCl solution
C-2	0.5	100	0	0	3.5% NaCl solution	3.5% NaCl solution
CC5-2	0.5	95	5	0	3.5% NaCl solution	3.5% NaCl solution
CC10-2	0.5	90	10	0	3.5% NaCl solution	3.5% NaCl solution
CC15-2	0.5	85	15	0	3.5% NaCl solution	3.5% NaCl solution
CS5-2	0.5	95	0	5	3.5% NaCl solution	3.5% NaCl solution
CS10-2	0.5	90	0	10	3.5% NaCl solution	3.5% NaCl solution

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
