# Peer review of "Hydration and Mechanical Properties of Calcium Sulphoaluminate Cement Containing Calcium Carbonate and Gypsum under NaCl Solutions"

_materials, 2022, doi:10.3390/ma15030816_

Round 1

Reviewer 1 Report

Dear authors,

The article is methodologically very good. I recommend it for approval - but I would like you to answer a few questions and make a few clarifying changes to the article:

- What do you think the $ sign means, is it not a mistake?
- The - above the 'S' stands for free electron pair or what?
- When describing XRD you did not specify the type of tube or wavelength, please correct this,
- The addition of NaCL is generally known to accelerate the setting of gypsum, increase the ego-thermal effect and increase the brittleness of concrete, describe something more in the intro because this is missing.

Having taken these comments into account, I recommend your article for acceptance. It was seriously fun to read. Keep up the good work.

Best regards

Reviewer

Author Response

Dear Reviewers,

Thank you for your letter and for the reviewers’ comments concerning our manuscript entitled “Hydration and mechanical properties of calcium sulphoaluminate cement containing calcium carbonate and gypsum under NaCl solutions”. (ID: materials - 1509971). On behalf of my co-authors, we thank you very much for giving us an opportunity to revise our manuscript. Those comments are all valuable and very helpful for revising and improving our paper, as well as the important guiding significance to our researches. We have studied comments carefully and have made correction which we hope meet with approval. The main corrections in the paper and the responds to the reviewer’s comments are as flowing:

Responds to the reviewer’s comments:

Reviewer #1:

  1. Response to comment: What do you think the $ sign means, is it not a mistake?

Response for 1: Thanks for your careful work. I'm sorry that S(_) and $ confuse you. $ was a mistake. S(_) refers to SO3 and is just the abbreviation of SO3. Now I have changed all the C4A3$ into C4A3S(_).

  1. Response to comment: The - above the 'S' stands for free electron pair or what?

Response for 2: Thanks for your careful work. I'm sorry that S(_) and $ confuse you. $ was a mistake. S(_) refers to SO3 and is just the abbreviation of SO3. Now I have changed all the C4A3$ into C4A3S(_).

  1. Response to comment: When describing XRD you did not specify the type of tube or wavelength, please correct this.

Response: Thanks for your careful work. We are so sorry that we did not specify the type of tube or wavelength. XRD analysis was performed by using Rigaku SmartLab 3000A diffractometer with Cu Ka radiation.

  1. Response to comment: The addition of NaCL is generally known to accelerate the setting of gypsum, increase the ego-thermal effect and increase the brittleness of concrete, describe something more in the intro because this is missing.

Response: Thanks for your careful work. We are very sorry that we neglected the influence of NaCl on cement hydration and performance in the introduction and we have added this in the introduction.

Having taken these comments into account, I recommend your article for acceptance. It was seriously fun to read. Keep up the good work.

Response: Thanks for your work.

Special thanks to you for your good comments.

Reviewer 2 Report

in my opinion this article is interesting.
The authors have studing the hydration mechanisms of sulphoaluminate cement containing calcium carbonate and gypsum under sodium chloride solutions. This work is interesting regarding the durability of cement-based materials.
- An adequate characterization of the material used is included regarding to Calcium sulphoaluminate cement, but the characterization of the rest of the materials used in the mixtures should be included, that is, the degree of purity of the gypsum used and calcium carbonate.
- The tested mixtures are adequately explained, it is suggested to use include several image in which it can be verified how the mixtures were made in the laboratory.
- In the results section, the authors include several figures, but some correlation between results should be included at the end of this section, for example between resistance to compression and shrinkage or the diameter of pore size.
- Finally in conclusions, several specific conclusions should be included of additionally. In my opinion, each specific conclusion must be indicated in each paragraph, and between 6 and 8 specific conclusions should be included. A final conclusion must be included in all the work.

Author Response

Dear Reviewers,

Thank you for your letter and for the reviewers’ comments concerning our manuscript entitled “Hydration and mechanical properties of calcium sulphoaluminate cement containing calcium carbonate and gypsum under NaCl solutions”. (ID: materials - 1509971). On behalf of my co-authors, we thank you very much for giving us an opportunity to revise our manuscript. Those comments are all valuable and very helpful for revising and improving our paper, as well as the important guiding significance to our researches. We have studied comments carefully and have made correction which we hope meet with approval. The main corrections in the paper and the responds to the reviewer’s comments are as flowing:

Responds to the reviewer’s comments:

Reviewer #2:

  1. Response to comment: An adequate characterization of the material used is included regarding to Calcium sulphoaluminate cement, but the characterization of the rest of the materials used in the mixtures should be included, that is, the degree of purity of the gypsum used and calcium carbonate.

Response: We are so sorry that we did not pay attention to the characterization of calcium carbonate and gypsum. We have added XRD analysis and purity description.

  1. Response to comment: The tested mixtures are adequately explained, it is suggested to use include several image in which it can be verified how the mixtures were made in the laboratory.

Response: We are sorry that we did not clearly express the process of making the mixture in the laboratory, we have added a few pictures to illustrate the process.

  1. Response to comment: In the results section, the authors include several figures, but some correlation between results should be included at the end of this section, for example between resistance to compression and shrinkage or the diameter of pore size.

Response: We are very sorry for ignoring the correlation between the results when writing. Now we have added this description.

  1. Response to comment: Finally in conclusions, several specific conclusions should be included of additionally. In my opinion, each specific conclusion must be indicated in each paragraph, and between 6 and 8 specific conclusions should be included. A final conclusion must be included in all the work.

Response: We are so sorry for the messy writing in conclusions. We have re-written this part according to the Reviewer’s suggestion.

Special thanks to you for your good comments.

Round 2

Reviewer 2 Report

The authors have modified all the suggestions presented by the reviewers and, in my opinion, this article can be accepted for publication.